# Treatment Options in Late-Line Colorectal Cancer: Lessons Learned from Recent Randomized Studies

**DOI:** 10.3390/cancers16010126

**Published:** 2023-12-26

**Authors:** Line Schmidt Tarpgaard, Stine Brændegaard Winther, Per Pfeiffer

**Affiliations:** 1Department of Oncology, Odense University Hospital, 5000 Odense C, Denmark; stine.winther@rsyd.dk (S.B.W.); per.pfeiffer@rsyd.dk (P.P.); 2Department of Clinical Research, University of Southern Denmark, 5230 Odense M, Denmark

**Keywords:** colorectal cancer, metastatic, refractory, molecular characterization, biomarker-driven strategies, targeted agents

## Abstract

**Simple Summary:**

Advancements in treating patients with metastatic colorectal cancer have shown remarkable progress in the last two decades. Enhanced comprehension of tumor biology via molecular profiling has broadened treatment avenues. The approach to treating patients with metastatic colorectal cancer has evolved from a uniform method to a more individualized one. It’s now clear that colorectal cancer manifests in diverse forms, characterized by varied molecular subtypes and genetic mutations, demanding personalized treatment approaches. This review delves into the latest clinical findings concerning late-stage treatment options for patients with metastatic colorectal cancer, mainly focusing on randomized trials wherever available. We include recommendations for options in unselected patients and therapies that should only be offered in patients with distinct tumor profiles.

**Abstract:**

Systemic treatment of metastatic colorectal cancer (mCRC) has improved considerably over the past 20 years. First- and second-line combinations of 5FU, oxaliplatin, and irinotecan, with or without anti-angiogenic and/or anti-EGFR antibodies, were approved shortly after the turn of the millennium. Further triumphs were not seen for almost 10 years, until the approval of initially regorafenib and shortly after trifluridine/tipiracil. A growing understanding of tumor biology through molecular profiling has led to further treatment options. Here, we review the most recent clinical data for late-line treatment options in mCRC, focusing on randomized trials if available. We include recommendations for options in unselected patients and therapies that should only be offered in patients with distinct tumor profiles (e.g., BRAF mutations, KRAS G12C mutations, HER2 amplification, deficient MMR, or NTRK gene fusions).

## 1. Introduction

Colorectal cancer (CRC) stands as the third most prevalent cancer globally, with more than 1.9 million new diagnoses recorded every year [1]. Despite advances in early detection and treatment strategies, nearly half of these patients are faced with metastatic CRC (mCRC), either at their initial diagnosis or later, owing to disease recurrence or progression. Sadly, more than 0.9 million individuals, accounting for almost 50% of CRC patients, succumb to this condition annually, presenting a formidable challenge for the field of oncology [1].

Since the dawn of the new millennium, the therapeutic landscape for mCRC has undergone significant transformations. The standard approach now involves doublet or triplet chemotherapy alongside targeted agents [2,3]. In the contemporary treatment strategy for most mCRC patients, a series of sequential systemic therapies are administered. Nevertheless, with each treatment course, 20–50% of patients do not qualify for further therapeutic interventions. This percentage greatly hinges on patient selection, as individuals enrolled in clinical trials tend to receive additional therapy lines more frequently than those in unselected cohorts. The European Society for Medical Oncology (ESMO) has communicated a treatment objective wherein 50% of ‘fit’ first-line patients should be eligible for third-line therapy [2]. Regrettably, in real-world settings, only 25–30% of ‘real-world first-line patients’ receive third-line therapy, and merely 10–15% obtain fourth-line treatments [4,5].

First- and second-line chemotherapy typically combine 5-fluorouracil (5FU) and irinotecan, followed by 5FU and oxaliplatin upon disease progression (or vice versa) [6]. Often, molecular-targeted medications like epidermal growth factor receptor (EGFR) inhibitors (for patients with RAS/BRAF wild-type tumors) or anti-angiogenic drugs are included, applicable to all patients irrespective of their molecular subtypes [2,3,7]. The mCRC treatment has shifted from a ‘one-size-fits-all’ strategy to a more personalized one. It is now evident that CRC is an immensely diverse disease, with numerous molecular subtypes and genetic mutations. This diversity often necessitates tailored treatment strategies [2,3,8].

We review clinical data for late-line treatment options in patients with mCRC, including treatment options in unselected patients and therapies that should only be offered in patients with distinct tumor profiles, e.g., patients harboring BRAF mutations, KRAS G12C mutations, HER2 amplification, deficient MMR, or NRTK fusions, with a focus on randomized trials when available. The emergence of chemo-refractory CRC represents a formidable clinical challenge, and we provide a comprehensive review of the evolving landscape of therapy for chemo-refractory CRC and the promising new approaches that offer renewed hope to these patients. In this paper, we define chemo-refractory mCRC as patients that have received (and often progressed to) 5FU, irinotecan, oxaliplatin, and anti-EGFR (RAS and BRAF wild-type). In some randomized trials, prior anti-angiogenic therapy was not mandatory, but the majority of chemo-refractory patients have also been exposed to angiogenic therapy as first- and/or second-line therapy. We will explore treatment options for unselected patients and highlight the potential for precision medicine. 

## 2. Current Systemic Treatment beyond Second-Line

### 2.1. Unselected Patients with Chemo-Refractory mCRC

What is the expected outcome if chemo-refractory mCRC patients are not offered active therapy? Numerous randomized trials have tested a new drug or combination against best supportive care (BSC) in unselected patients with chemo-refractory mCRC and constantly revealed a progression-free survival (PFS) of around 1.5 months and an overall survival (OS) of 5 months (4–6) months for patients receiving BSC. First and second-line combinations of 5FU, oxaliplatin, and irinotecan, with or without targeted therapy, have been the standard since the turn of the millennium, but further triumphs have not been seen for almost 10 years. In a systematic review, it was concluded that conventional chemotherapeutic agents had limited or no activity as salvage therapy in chemo-refractory mCRC [9]. This lack of new active drugs lasted until the approval of regorafenib in 2012, shortly after the approval of trifluridine/tipiracil (FTD/TPI).

Several trials, such as CORRECT [10], RECOURSE [11], and SUNLIGHT [12], significantly improved outcomes in unselected patients. A prolonged PFS and OS in these pivotal randomized trials formed the basis for the approval of new drugs. Efficacy data from these randomized trials and the recently published FRESCO-2 trial [13] are summarized in Table 1, along with efficacy data from precision therapy in selected subgroups—mainly data from small phase 2 studies. 

Table 2 describes randomized studies that showed a significantly prolonged OS (and PFS) in unselected patients with chemo-refractory mCRC. Regorafenib is an oral tyrosine kinase inhibitor (TKI) targeting multiple pathways and exerting its anti-angiogenic effects after being internalized by cells, binding to and inhibiting the kinase domain of various receptors involved in angiogenesis. Unlike monoclonal antibodies, and despite numerous studies involving thousands of patients, no randomized study has demonstrated a survival advantage when combining chemotherapy with a TKI as a first- or second-line therapy in mCRC [7]. Thus far, regorafenib is the only TKI integrated into the treatment sequence for mCRC patients. It gained approval from the U.S. Food and Drug Administration (FDA) and the European Medicines Agency (EMA) in 2012 and 2013, respectively, following positive results from the CORRECT trial [10]. The CORRECT trial, a randomized, double-blind phase 3 study, investigated the efficacy of regorafenib at a daily dosage of 160 mg for 21 days in 28-day cycles compared to a placebo in 760 patients with mCRC previously treated with standard therapies [10]. The study successfully met its primary endpoint by showing a significant increase in median OS with 1.4 months; 6.4 months in the regorafenib group versus 5.0 months in the placebo group (HR 0.77; *p* = 0.005); and also improvement in median PFS (HR 0.49). The effectiveness of regorafenib was confirmed in the Asian CONCUR study, which showed prolonged median OS from 6.3 to 8.8 months [14].

More than 90% of patients in the regorafenib arm experienced treatment-related adverse events (TRAEs) of any grade. Hand-foot reactions, hypertension, fatigue, diarrhea, and laboratory abnormalities were the most common TRAEs. Additionally, more than 60% of patients treated with regorafenib required dose reductions, especially within the first two cycles. However, discontinuing treatment due to adverse events was uncommon, and quality of life (QoL) did not get worse [10].

Three studies have investigated FTD/TPI as monotherapy versus placebo [11,15,16]. FTD/TPI is an oral fluoropyrimidine consisting of two compounds: trifluridine, a cytotoxic nucleic acid analogue, and tipiracil, a thymidine phosphorylase inhibitor that blocks the trifluridine enzymatic degradation. FTD/TPI was approved by the FDA and the EMA in 2015 and 2016, respectively, following the positive results of the RECOURSE trial. The primary endpoint (OS) was 7.1 months in the FTD/TPI group and 5.3 months in the placebo group (HR 0.68; *p* < 0.001), and PFS was also significantly prolonged (HR 0.48; *p* < 0.001) [11].

A Danish randomized trial demonstrated further improvement with a median survival advantage of 2.7 months (6.7 to 9.4 months; HR 0.55) when combining bevacizumab with FTD/TPI compared to FTD/TPI monotherapy. This benefit was consistent across various patient subgroups, even those who had previously received bevacizumab in their immediate prior therapy line [17]. These encouraging findings were recently corroborated by data from the SUNLIGHT study [12].

**Table 2 cancers-16-00126-t002:** Principal randomized trials that showed a significant survival benefit in unselected patients with chemo-refractory mCRC.

Author, Year [Ref.] Trial Name	Regimen	N	ORR %	*p*	PFS Months	HR 95%CI	OS Months	HR 95%CI	QoL
Grothey, 2013 [10] CORRECT	Placebo	255	0	NS	1.7	0.49 0.42–0.58	5.0	0.77 0.64–0.94	→
	Regorafenib	505	1		1.9 *		6.4 *		
Li, 2014 [14] CONCUR	Placebo	68	0	0.045	1.7	0.31 0.22–0.44	6.3	0.55 0.40–0.77	→
	Regorafenib	136	4 *		3.2 *		8.8 *		
Yoshino, 2012 [15]	Placebo	57	0	NS	1.0	0.41 0.28–0.59	6.6	0.56 0.53–0.81	ND
	FTD/TPI	112	1		2.0 *		9.0 *		
Mayer, 2015 [11] RECOURSE	Placebo	266	0	0.045	1.7	0.48 0.41–0.57	5.3	0.68 0.58–0.81	ND
	FTD/TPI	534	2 *		2.0 *		7.1 *		
Xu, 2018 [16] TERRA	Placebo	135	0	NS	1.8	0.43 0.34–0.54	7.1	0.79 0.62–0.99	ND
	FTD/TPI	271	1		2.0 *		7.8 *		
Li, 2013 [18] FRESCO	Placebo	138	0	0.01	1.8	0.26 0.21–0.34	6.6	0.65 0.51–0.83	→
	Fruquintinib	278	5 *		3.7 *		9.3 *		
Dasari, 2023 [12] FRESCO-2 (4 L)	Placebo	229	0	0.06	1.8	0.32 0.27–0.39	4.8	0.66 0.55–0.80	Ongoing
	Fruquintinib	458	2		3.7 *		7.4 *		
Active therapy as comparator									
Pfeiffer, 2020 [17] Danish randomized phase 2	FTD/TPI	47	0	NS	2.6	0.45 0.29–0.72	6.7	0.55 0.32–0.94	ND¤
	FTD/TPI + bevacizumab	46	2		4.6 *		9.4 *		
Prager, 2023 [13] SUNLIGHT	FTD/TPI	246	1	<0.05	2.4	0.44 0.36–0.54	7.5	0.61 0.49–0.77	Ongoing
	FTD/TPI + bevacizumab	246	6		5.6 *		10.8 *		

Abbreviations: Ref. = reference, N = number of patients, ORR = response rate, PFS = progression-free survival, HR = hazard ratio, OS = overall survival, CI = confidence interval, QoL = quality of life; (→ = no difference in QoL; NS = non-significant; ND = not done or not reported; * = significant difference; FTD/TPI = trifluridine/tipiracil. ND¤: Formally QoL not evaluated but longer time to worsen the performance status.

SUNLIGHT was an international, randomized phase 3 study encompassing 492 patients with chemo-refractory mCRC. The combination of bevacizumab with FTD/TPI extended the median OS by 3.3 months (from 7.5 to 10.8 months, HR 0.61; *p* < 0.001) and PFS by 3.2 months (HR 0.44; *p* < 0.001). The combination of bevacizumab with FTD/TPI was well tolerated, with hematological toxicities, especially neutropenia (43%), being the most frequently reported severe adverse events.

In the two trials testing FTD/TPI with bevacizumab, a formal QoL evaluation was not reported, but in both trials, a longer time to worsening of the performance status was observed [12,17]. 

On 22 June 2023, EMA granted a positive opinion, recommending a modification to the marketing authorization terms for FTD/TPI. The revised indication states that FTD/TPI, in combination with bevacizumab, is indicated for the treatment of adult patients with mCRC who have undergone two prior anticancer treatment regimens, including fluoropyrimidine-based, oxaliplatin-based, and irinotecan-based chemotherapies, as well as anti-VEGF agents and/or anti-EGFR agents. Similarly, on 2 August 2023, FDA approved FTD/TPI in conjunction with bevacizumab for the same indication as mentioned above. In patients with no specific targets, we suggest that they primarily be offered FTD/TPI and bevacizumab instead of regorafinib due to the difference in side effects between the two regimes if the treatment is authorized by the country’s authorities.

Fruquintinib is a highly selective and potent oral inhibitor of VEGFR-1, -2, and -3, pivotal in impeding tumor angiogenesis [13]. It was specifically designed to enhance kinase selectivity, aiming to reduce off-target toxicities, enhance tolerability, and ensure consistent target coverage. Patients have generally exhibited good tolerance to fruquintinib, and it is also under investigation in combination with other anti-cancer therapies. Based on data from the FRESCO trial [17], fruquintinib has received approval in China. In the FRESCO trial, fruquintinib treatment extended the median OS by 2.7 months (from 6.6 to 9.3 months, HR 0.65; *p* < 0.001). FRESCO-2, on the other hand, was an international, randomized, double-blind, placebo-controlled, phase 3 study conducted across 14 countries in 124 hospitals [13]. In less than 16 months, 691 patients with mCRC were enrolled. These patients had previously undergone all currently approved standard therapies (5FU, oxaliplatin, irinotecan, anti-EGFR-only for RAS wild-type, and bevacizumab), and they should have progressed on or been intolerant to FTD/TPI regorafenib, or both. FRESCO-2 thus included patients who were more heavily pretreated than those in the SUNLIGHT study.

Fruquintinib significantly prolonged the median OS (primary endpoint) by 2.6 months (from 4.8 to 7.4 months, HR 0.66, *p* < 0.0001) and PFS from 1.8 months to 3.7 months (HR 0.32, *p* < 0.0001). The benefits of fruquintinib were observed across all patient subgroups, including those with prior treatment involving FTD/TPI or regorafenib. Among the most commonly reported grade ≥3 adverse events were hypertension (14% versus 1%), asthenia (8% versus 4%), and hand–foot syndrome (6% versus 0%). The proportion of patients discontinuing treatment due to adverse events was similar between the fruquintinib and placebo groups (20% vs. 21%).

As a third-line therapy in unselected patients, the combination of FTD/TPI with bevacizumab is poised to become the new standard of care in mCRC. Fruquintinib will likely emerge as the standard of care for patients progressing to FTD/TPI and bevacizumab, even though only a few patients in the FRESCO-2 study had received the FTD/TPI and bevacizumab combination immediately before starting fruquintinib. While all subgroups in the FRESCO-2 study seemed to benefit from fruquintinib, it remains intriguing to determine if patients who received bevacizumab in the immediate prior line before starting fruquintinib will experience the same extent of benefits from this promising new anti-angiogenic therapy.

On 8 November 2023, fruquintinib received approval from the FDA for treating patients with chemo-refractory mCRC, and the EMA has validated and accepted for regulatory review the marketing authorization application (MAA) for fruquintinib.

### 2.2. Randomized Trials That Did Not Show a Prolonged Survival in Late-Line Treatment of Patients with mCRC

Several randomized trials [19,20,21,22,23,24,25] have explored other new treatment options compared with BSC in unselected patients with chemo-refractory mCRC but failed to show a prolonged OS or improved QoL (Table 3). No study produced a clinically meaningful response rate (RR), whereas three studies showed a significantly improved PFS compared to BSC. In the LUME trial [19] that evaluated nintedanib, a triple antiokinase inhibitor, a modest but significant increase in PFS was shown; in the ALTER0703 trial [21], anlotinib (a TKI inhibitor targeting VEGFR 1–3, PDGFR α/β, and stem cell factor receptor) increased PFS from 1.5 to 4.1 months; and finally, Xu et al. [22] showed that famitinib, a tyrosine kinase inhibitor targeting VEGFR 2–3, PDGFR, FLT3, and RET, increased PFS from 1.5 to 2.8 months. None of the drugs summarized in Table 3 have been approved.

A number of randomized trials have compared two or three active drugs, combinations, or new treatment strategies in patients with chemo-refractory mCRC (Table 4), but no trial could demonstrate a prolonged OS [26,27,28,29,30,31,32,33,34,35]. Due to adverse events, especially skin toxicity and fatigue, a commonly used strategy for the upstart of regorafenib is to initiate therapy with a reduced dose and then escalate the dose if no severe toxicity is seen after 2 or 4 weeks of therapy. In the REDOS trial, PFS and OS were numerically but not significantly longer, but probably a valid conclusion is that a reduced dose with escalation is a safe strategy [26]. 

The addition of ruxolitinib (an oral selective JAK1/2 inhibitor) to regorafenib did not improve efficacy [28]. Priming with RRx-001 (a cysteine-targeted alkylating agent that yields nitric oxide upon selective alkylation of cysteine moieties) before treatment with irinotecan prolonged PFS but not RR or OS when compared to regorafenib, and due to the small number of patients (patients were randomized 1:2), the results can at most only be hypothesis-generating [32]. 

A study of sequential strategy explored whether treatment with regorafenib should be followed by cetuximab or vice versa in patients with KRAS wild-type tumors and showed divergent results. RR was significantly increased with a tendency toward increased PFS if cetuximab was administered before regorafenib, but in contrast, OS was significantly shorter [29]. In a similar selected population of patients with KRAS wild-type tumors, the addition of brivanib (a tyrosine kinase inhibitor targeting vascular endothelial growth factor receptor and fibroblast growth factor receptor) to cetuximab resulted in increased RR and PFS but no OS benefit, and QoL was actually inferior [30]. FTD/TPI is, in general, very well tolerated, with neutropenia as the main adverse event. Therefore, it was obvious that FTD/TPI should be combined with panitumumab. In the VELO trial [33,34], the combination of FTD/TPI plus panitumumab increased the RR numerically. It prolonged significantly PFS (primary endpoint), but with no impact on OS, compared to FTD/TPI monotherapy, neither in the total population nor in the subgroup with baseline circulating tumor DNA (ctDNA) RAS/BRAF wild-type. 

Finally, in RAS-mutated patients, a trial of irinotecan vs. sorafenib vs. irinotecan plus sorafenib showed no difference in efficacy data or QoL [35].

## 3. Immunotherapy in Later-Lines

Based upon the Keynote 177 trial [36], immunotherapy is standard as first-line therapy in patients with deficient DNA mismatch repair/microsatellite instability (dMMR)/(MSI) CRC. If a patient, for some reason, did not receive upfront prior immunotherapy, immunotherapy should be offered as the preferred regimen, also in later-lines, if available [2,3]. In contrast, no trial has shown a clinically meaningful benefit of immunotherapy, as monotherapy or in combination, in patients without dMMR (proficient MMR). However, the search for effective regimens continues, and so far, three randomized trials have shown unsatisfactory results. A Canadian randomized phase 2 trial assessed durvalumab + tremelimumab with placebo in 180 patients with chemo-refractory mCRC but found no difference in RR and PFS but a slightly longer OS (6.6 months vs. 4.1 months) [37]. Only two patients had dMMR/MSI. Mettu et al. tested the addition of atezolizumab to capecitabine and bevacizumab in 128 patients with refractory dMMR mCRC and found no efficacy on RR and OS, but a minor improvement in PFS (3.6 months vs. 4.6 months) in favor of atezolizumab [38]. However, the authors concluded that the improvement provided limited, not clinically meaningful, benefit. In the IMblaze370 study, atezolizumab plus cobimetinib or atezolizumab monotherapy versus regorafenib was compared in the third-line setting in 273 patients with mCRC, with only 2% of the patients having MSI-tumors [39]. Primary endpoint OS was not met (7.1 months vs. 8.9 months vs. 8.5 months), and there was no significant difference in PFS and RR and a tendency toward more grade 3 toxicity among patients who received atezolizumab plus cobimetinib. 

Thus, based on current evidence, immunotherapy can only be recommended for patients with dMMR/MSI. 

## 4. Selected Subgroup of Patients with Chemo-Refractory mCRC

Knowledge of the molecular biological profile is increasingly important as new targeted agents are constantly being introduced in patients with chemo-refractory mCRC. Below, we summarize the most promising drugs or combinations that are suitable for subgroups of patients with specific molecular and biological characteristics.

## 5. Re-Challenge with Anti-EGFR

Patients with RAS and BRAF wild-type tumors will frequently be offered anti-EGFR therapy in the early stages, but despite the benefit of anti-EGFR therapy in patients with RAS wild-type and BRAF wild-type mCRC, most of the patients will develop resistance within 12 months of treatment [40]. Various theories have been suggested to describe the mechanism of this secondary resistance. Diaz et al. [41] showed that some of the patients with RAS wild-type tumors who received cetuximab later developed RAS mutations during treatment. Based on mathematical models, it was suggested that the secondary resistance was caused by a subpopulation of mutated clones expanding under selective treatment pressure. Further investigations examined tumor genotyping with liquid biopsy and demonstrated a decline in mutated clones upon EGFR blockade withdrawal [40,42]. Based on these studies, patients who initially responded to anti-EGFR therapy but later progressed could possibly benefit from re-treatment with anti-EGFR therapy. 

Even though the evidence for re-challenge is low, this option is promising if tumor shrinkage is wanted. The CRICKET trial found a median PFS of 6.6 months for patients re-challenged with cetuximab and irinotecan (CetIri) and a very impressive RR of 54% [43]. In a small phase 2 trial of 28 patients, Cremolini et al. found a RR of 21% with CetIri [44]. Approximately 12 of 25 patients (48%) with RAS wild-type ctDNA had significantly longer PFS than those with RAS mutated ctDNA (median PFS 4.0 vs. 1.9 months), and patients who achieved partial response had no ctDNA RAS mutations. The JACCRO CC-08 trial may have been an outlier when it comes to tumor shrinkage because only one of 34 patients (3%) qualified for a response when re-challenged with CetIri in KRAS wild-type mCRC patients [45]. Masuishi et al. also tested the importance of the interval from primary anti-EGFR therapy to re-challenge and found that a long-term interval correlated with a better outcome. In the CHRONOS trial, the RR was 30% and the PFS was 3.7 months [46]. 

We suggest that patients with initial RAS and BRAF wild-type tumors be screened for newly acquired RAS and BRAF mutations in liquid biopsies. If no mutations have been acquired, rechallenge with anti-EGFR is thus a relevant option for patients who have responded to previous anti-EGFR therapy.

## 6. Patients with KRAS G12C Mutations

Almost half of patients with CRC exhibit KRAS mutations, but the KRAS G12C mutation appears in only 3%. Novel insights into the structure and biochemical properties of mutant KRAS G12C have led to the development of inhibitors like sotorasib and adagrasib [47]. Pre-clinical studies have shown that inhibition of KRAS G12C rapidly reactivates the EGFR-mediated MAPK pathway, but inhibition of EGFR increased efficacy and maintained response for a longer period. In the non-randomized KRYSTAL-1, ORR was much higher (46%) with a combination of adagrasib and cetuximab [48]. 

Considering that the KRAS G12C mutation is present in only 3% of mCRC, it is very impressive that 160 patients were included in the randomized CodeBreaK300. Patients were randomized to two different doses of sotorasib (960 mg or 240 mg) in combination with panitumumab versus standard of care (SOC) (FTD/TPI or regorafinib) as second- or later-line therapy [49]. Median PFS in the high-dose group was prolonged from 2.2 months to 5.6 months. OS data are not yet mature, but preliminary data showed no major difference in outcome between sotorasib–panitumumab and SOC. Overall RR was 26%, 6%, and 0% in the 960 mg sotorasib–panitumumab, 240 mg sotorasib–panitumumab, and SOC groups, respectively. Skin-related toxic effects and hypomagnesemia were the most common adverse events observed with sotorasib–panitumumab.

In patients with KRAS G12C mutated tumors, treatment with a KRAS G12C inhibitor in combination with anti-EGFR therapy should be pursued, if possible, in clinical trials to ensure further evidence. 

## 7. BRAF

Nowadays, patients with BRAF V600E mutations are most often treated with encorafenib and cetuximab as second-line therapy based on the BEACON trial [50], and patients with dMMR should preferably be treated with pembrolizumab (or other kinds of immunotherapy) in the first-line setting based on the Keynote 177 trial [36].

Many BRAF-mutations are identified, but the most common is BRAFV600E, where thymidine is replaced with adenine at nucleotide 1799, resulting in an amino acid change from valine (V) to glutamine (E) in the BRAF protein. This mutation constitutes approximately 90% of all BRAF mutations in CRC and occurs in 5–21% of patients with mCRC [51]. BRAFV600E mutated tumors are more likely to have mucinous histology, low differentiation, and are associated with dMMR, with approximately 25% having simultaneous dMMR. Conversely, about 5% of mCRC patients have dMMR, and roughly 34% are BRAFV600E-mutated [52]. 

Patients with BRAFV600E-mutated mCRC have a worse prognosis than those without the BRAFV600E mutation, even after curative-intent metastasis surgery.

The BEACON trial [50] included 665 patients with BRAFV600E mutated, RAS wild-type mCRC who had previously received first- or second-line chemotherapy and were randomized to triplet therapy (encorafenib, binimetinib, and cetuximab), doublet therapy (encorafenib and cetuximab), or standard treatment (irinotecan/cetuximab or FOLFIRI/cetuximab). Patients randomized to triplet or doublet BRAF inhibition showed improved OS compared to the control group (9.0 months vs. 8.4 months vs. 5.4 months). This improvement in efficacy was confirmed in a later update [50,53]. The study was not powered to compare doublet and triplet BRAF inhibition, but the two treatments demonstrated similar effects on OS and PFS. Furthermore, there was a slightly higher incidence of grade 3 toxicity with triplet and control treatments compared to doublet treatment, which led to doublet treatment being approved as the standard of care in patients with BRAFV600E-mutated mCRC.

Thus, evidence is already well established, and patients with BRAFV600E-mutated mCRC should receive second-line encorafenib and cetuximab as SOC.

## 8. HER2 

HER2, a member of the EGFR family, is typically activated through ligand binding and dimerization with other EGF family receptors. HER2 overexpression, often due to ERBB2 gene amplification, activates replication signals independently of ligand-bound dimerization partners. In mCRC, HER2 overexpression varies, with the highest prevalence in RAS wild-type and rectal cancer tumors (approximately 5–8%). HER2 positivity was not a strong factor when mCRC patients were treated with chemotherapy alone, but HER2 positivity predicted worse ORR and PFS when receiving anti-EGFR therapy compared with chemotherapy alone [54,55].

Table 5 summarizes studies, primarily phase 2, investigating the efficacy of HER2 targeted therapy, often dual targeted therapy, in patients with HER2 positive (HER2+) mCRC [56,57,58,59,60,61,62,63,64,65]. Patients with HER2+ mCRC benefit from anti-HER2 treatment, with RR ranging from 10 to 55% and PFS and OS outcomes spanning 2.9–6.9 months and 10.6–24.1 months, respectively. 

The HERACLES study, a phase 2 trial, demonstrated a 30% ORR with dual HER2 inhibition in treatment-refractory patients [56]. MyPathway, a basket trial, confirmed the efficacy of pertuzumab and trastuzumab in HER2-amplified mCRC, achieving a 32% ORR [58].

Tucatinib, an anti-HER2 oral treatment, in combination with trastuzumab showed promise in the MOUNTAINEER trial, with a 55% ORR [64]. Trastuzumab deruxtecan, an antibody-drug conjugate, exhibited a 45.3% ORR in HER2-overexpressing mCRC patients in the DESTINY-CRC01 trial. However, interstitial lung disease was observed in 9.3% of patients [62,63].

We recommend dual targeted anti-HER2 therapy in patients with HER2-positive chemo-refractory mCRC if regression is the aim, preferably in clinical trials.

## 9. TRK

Two drugs (larotrectinib and entretinib) have been approved by the FDA and EMA for the treatment of patients with an NTRK gene fusion in solid tumors. Neutrotrophic tyrosine receptor kinase (NRTK) genes encode tropomyosin receptor kinase (TRK) proteins, and rearrangements can result in somatic NTRK gene fusions, which can cause the uncontrolled growth of tumors [66]. NTRK gene fusions can be found in only 0.2% of unselected CRC, but with a higher prevalence in patients with RAS/BRAF wild-type and dMMR tumors. Data presented at the ESMO GI Congress 2022 provided updated efficacy information in CRC patients for larotrectinib (19 patients) and entrectinib (10 patients) with RR of 47% and 20%, PFS of 5.5 months and 3.0 months, and OS of 12 and 16 months, respectively [67,68]. 

For patients with NRTK gene fusion, we recommend targeted treatment as mentioned above in clinical trials. 

## 10. Discussion

As just reviewed, CRC has become a more complex and multifaceted disease, where tumor profiling is increasingly crucial for treatment selection both in early and later-line therapy. 

Less than one-third of ‘real-world first-line patients’ are candidates for third- or later-line therapy [4,5]. When dealing with a fit patient with refractory mCRC, there are several points that should be addressed. What prior therapy did the patient receive, what was the duration of therapy and response to treatment, how did the patient tolerate prior therapy, are there any persistent long-term toxicity, and the patient’s attitude toward further therapy are also important. However, often the most important point is the molecular status of the tumor. If the tumor is left-sided, RAS, or BRAF wild-type, you should check that the patient did receive prior anti-EGFR therapy and that patients with dMMR or BRAF V600E mutated tumors have been exposed to immunotherapy and BRAF inhibitors, respectively. If not, the patient should, of course, be offered the relevant treatment. 

In the ESMO 2022 guidelines [3], regorafenib and FTD/TPI were mainly recommended for chemo-refractory patients with RAS mutations; however, a recent update [69] recommended regorafenib and FTD/TPI for all chemo-refractory patients and added FTD/TPI-bevacizumab as an option for unselected patients. These drugs are recommended and approved based on prolonged PFS and OS, but regression is seldom obtained, and the primary aim with these drugs is thus to maintain well-being for as long as possible. Similar considerations are valid for the benefit of fruquintinib.

However, when the aim of therapy is tumor shrinkage, it is prudent to explore other targeted options, even though the scientific evidence for more targeted therapy is lower (Table 1). However, definitive regression of the tumor burden was seldom achieved, and therefore other options, even though the scientific evidence is lower, might be the preferred option for patients with distinct tumors and a demand for shrinkage. Definitive regression is often the primary goal in patients with symptomatic disease, if conversion is needed before local curative therapy, and probably in patients with massive organ involvement. If possible, these patients should be included in prospective trials. Outside clinical trials, a number of biological features (therapy for patients with RAS wild-type, HER2 positivity, BRAF V600E mutation, KRAS G12C mutation, NTRK gene fusions, and dMMR) are well documented, but the number of targets continues to increase and can make a patient a candidate for precision medicine if it is available in your country/region. RAS, BRAF V600E, and MMR/MSI status should be tested upfront. If this has not been conducted previously, we suggest that patients who are candidates for 3rd line treatment should be tested for HER2 positivity, NTRK gene fusions, and others for targeted treatment in clinical trials, if possible. 

Patients who are sustained RAS and BRAF V600E wild-type are candidates for re-challenge or re-introduction with anti-EGFR therapy, and even though data from randomized trials are lacking, we recommend combining anti-EGFR therapy with irinotecan [44,45].

Patients with HER2-positive tumors are candidates for (often dual) anti-HER2 therapy, and patients with rare NRTK fusions should be offered entrectinib or larotrectinib. Lastly, patients with KRAS G12C mutations should be evaluated for combination therapy with KRAS G12C inhibitors and anti-EGFR therapy. A response rate of at least 20%, a prolonged median PFS of 5 months with improved or preserved QoL, and a longer median OS of sometimes 12 months or more are what can be expected. In addition, many patients are candidates for more than one treatment attempt for refractory mCRC.

Overall, the late-line treatment options for patients with mCRC are an ongoing challenge; however, evidence gathers around more treatment options, some led by selected targets and others relevant to the unselected population.

## 11. Conclusions

Systemic treatment of patients with mCRC has improved considerably over the past 20 years, treatment strategy has shifted from a ‘one-size-fits-all’ strategy to a more personalized one. It is now evident that CRC is an immensely diverse disease, with numerous molecular subtypes and genetic mutations. This diversity often necessitates tailored treatment strategies. There are now several late-line treatment options, both for patients with genetic alterations and for those without. The most important thing for the patient is that they are offered all available drugs. This review has summarised the latest studies and provided recommendations for treatment choices. 

## Figures and Tables

**Table 1 cancers-16-00126-t001:** Summary of treatment options for patients with chemo-refractory mCRC. For comparison, a summary of outcome data for the best supportive care from randomized trials has been included.

Trial Name [Ref.]	% of mCRC	Phase	Treatment Options	N	ORR %	PFS Months	OS Months
No therapy, data from numerous randomized trials with BSC				>1000	0	1.5	5
Unselected patients, data from pivotal randomized trials							
CORRECT [10]	100%	3	Regorafenib	505	1	1.9	6.4
RECOURSE [11]	100%	3	FTD/TPI (TAS-102)	534	2	2.0	7.1
FRESCO-2 [12]	100%	3	Fruquintinib	458	2	3.7	7.4
SUNLIGHT [13]	100%	3	FTD/TPI + bevacizumab	246	6	5.6	10.8
Selected patients, data mainly from phase 2							
BRAFmut	8–12%	3	Encorafenib + cetuximab (2nd line)	220	20	4.2	8.4
HER2+ and RASwt	3–5%	2	Anti-HER2 treatment	19–53	10–55	2.9–6.9	10.6–24.1
RASwt and BRAFwt	30%	2	Anti-EGFR rechallenge	28–39	3–54	2.4–6.6	8.2–9.8
KRAS_G12C_	3%	1/2	Adagrasib and cetuximab	28	46	6.9	13.4
KRAS_G12C_	3%	3	Sotorasib and panitumumab	53	26	5.6	NR
NTRK gene fusions	<1%	2	Entrectinib, larotrectinib	10–19	20–47	3.0–5.5	12–16

Abbreviations: Ref. = reference, mCRC = metastatic colorectal cancer, N = number of patients, ORR = overall response rate, PFS = progression-free survival, OS = overall survival.

**Table 3 cancers-16-00126-t003:** Principal randomized trials that did not show prolonged survival in unselected patients with chemo-refractory mCRC. Trials comparing active therapy against the best supportive care/placebo.

Author, Year [Ref.] Trial Name	Regimen	N	ORR (%)	*p*	PFS (mo)	HR (95%CI)	OS (mo)	HR (95%CI)	QoL
Van Cutsem, 2018 [19] LUME	Placebo	382	0	NS	1.4	0.58 0.49–0.69	6.0	1.01 0.86–1.19	→
	Nintedanib	386	0		1.5 *		6.4		
Jonker, 2018 [20]	Placebo	144	0	NS	1.8	0.97 0.76–1.26	4.8	1.13 0.88–1·46	⇩
	Napabucasin	138	0		1.8		4.4		
Chi, 2021 [21] ALTER0703	Placebo	137	1	0.07	1.5	0.34 0.27–0.43	7.2	1.02 0.82–1.27	→
	Anlotinib	282	4		4.1 *		8.6		
Xu, 2017 [22]	Placebo	55	0	NS	1.5	0.60 0.41–0.86	7.2	NS	→
	Famitinib	99	2		2.8 *		7.4		
Grothey, 2018 [23]	Placebo	42	2	NS	1.9	1.13 0.76–1.67	6.1	1.43 0.93–2.19	ND
	Ontuxizumab	84	0		1.9		4.8		
Rao, 2004 [24]	Placebo	133	0	NS	2.6	1.22	6.1	NS	→
	Tipifarnib	235	1		2.6		5.7		
Caballero-Baños, 2016 [25]	Placebo	24	0	NS	2.3	NS	4.7	NS	ND
	ADC	28	0		2.7		6.2		

Abbreviations: Ref. = reference, N = number of patients, ORR = response rate, PFS = progression-free survival, HR = hazard ratio, OS = overall survival, CI = confidence interval, QoL = quality of life; (→ = no difference in QoL; ⇩= QoL worse); NS = non-significant; ND = not done or not reported; * = significant difference; ADC is an autolo-gous tumor lysate dendritic cell vaccine.

**Table 4 cancers-16-00126-t004:** Principal randomized trials that compared two or three active drugs, combinations, or new treatment strategies in patients with chemo-refractory mCRC.

Author, Year [Ref.] Trial Name	Regimen	N	ORR %	*p*	PFS Months	HR 95%CI	OS Months	HR 95%CI	QoL
Bekaii-Saab, 2019 [26] ReDOS	Regorafenib 160	62	-	-	2.0	0.84 0.57–1.24	6.0	0.72 0.47–1.10	→
	Regorafenib 80 to 160 mg	54	-		2.8		9.8		
Argiles, 2022 [27] REARRANGE	Regorafenib 160	101	2	NS	1.9	NS	7.4	NS	ND
	Regorafenib 120 to 160 mg	99	2		2.0		8.6		
	Regorafenib 160 1 w	99	3		2.0		7.1		
Fogelman, 2018 [28]	Regorafenib + placebo	111	5	NS	2.0	0.79 0.58–1.07	10.9	0.77 0.48–1.23	→
	Regorafenib + ruxolitinib	110	3		3.5		11.4		
Reid, 2023 [32] ROCKET	Regorafenib	10	0	NS	1.7	0.24 0.09–0.61	4.7	0.71 0.27–1.9	ND
	RRx-001 → irinotecan	24	21		6.1 *		8.6		
Shitara, 2019 [29] REVERCE	Regorafenib → cetuximab	51	4	0.03	2.4	0.97 0.62–1.54	17.4	0.61 0.39–0.96	→
	Cetuximab → regorafenib	50	20 *		4.2		11.6 *		
Siu, 2013 [30] AGITG CO.20 (KRASwt)	Cetuximab	374	7	0.004	3.4	0.72 0.62–0.84	8.1	0.88 0.74–1.03	⇩
	Cetuximab + brivanib	376	14 *		5.0 *		8.8		
Price, 2014 [31] ASPECCT	Cetuximab	500	20	NS	4.4	1.19 0.66–2.13	10.0	0.97 0.84–1.11	→
	Panitumumab	499	22		4.1		10.4		
Napolitano, 2023 [33,34] VELO	FTD/TPI	31	0	NS	2.5	0.48 0.28–0.82	13.1	0.96 0.54–1.71	ND
	FTD/TPI + panitumumab	31	10		4.0 *		11.6		
Samalin, 2020 [35] PRODIGE27 (RASmut)	Irinotecan	57	2	NS	1.9	NS	6.3	NS	→
	Sorafenib	57	2		2.1		5.6		
	Irinotecan + Sorafenib	59	4		3.6		7.2		

Abbreviations: Ref. = reference, N = number of patients, ORR = response rate, PFS = progression-free survival, HR = hazard ratio, OS = overall survival, CI = confidence interval, QoL = quality of life; (→ = no difference in QoL; ⇩= QoL worse); NS = non-significant; ND = not done or not reported; * = significant difference; FTD/TPI = trifluridine/tipiracil.

**Table 5 cancers-16-00126-t005:** Principal trials, mainly phase II, test anti-HER2 therapy in patients with chemo-refractory HER2+ mCRC.

Author, Year [Ref.] Trial Name	Treatment	N	Line	(K) RAS	ORR %	PFS Months	OS Months
Sartore-Bianchi, 2014 [56] Heracles-A	Trastuzumab + lapatinib	27	74% 4 L+	KRASwt	30	4.8	10.6
Sartore-Bianchi, 2020 [57] Heracles-B	Pertuzumab + TDM1	31	48% 4 L+	RASwt	10	4.1	-
Meric-Bernstam, 2019 [58] MyPathway	Trastuzumab + pertuzumab KRASwt KRASmut	57 43 13	67% 4 L+	23% RASmut	32 40 8	2.9 5.3 1.4	11.5 14.0 8.5
Nakamura, 2022 [59] Triumph	Trastuzumab + pertuzumab	27	78% 3 L+	RASwt	30	4.0	10.1
Gupta, 2020 [60] Tapur	Trastuzumab + pertuzumab	28	79% 3 L+	RASwt	14	4.0	58% 1Y
Chang, 2022 [61] HER2-FUSCC	Trastuzumab + pyrotinib	16	100% 2 L+	13% RASmut	50	7.5	16.8
Siena, 2021 [62,63] Destiny-CRC01	Trastuzumab-deruxtecan	53	Median 4	RASwt	45	6.9	15.5
Strickler, 2023 [64] Mountaineer	Trastuzumab + tucatinib	86	39% 3 L+	RASwt	38	8.2	24.1
Raghav, 2023 [65] Destiny-CRC02	T-DXd 5.4 mg/kg	82	Median 4	85% RASwt	38	5.8	13.4
	T-DXd 6.4 mg/kg	40	Median 4	85% RASwt	28	5.5	NR

Abbreviations: Ref. = reference, N = number of patients, ORR = response rate, PFS = progression-free survival, OS = overall survival, NR = not reached.

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
