# Peer review of "Treatment Options in Late-Line Colorectal Cancer: Lessons Learned from Recent Randomized Studies"

_cancers, 2023, doi:10.3390/cancers16010126_

Round 1

Reviewer 1 Report

Comments and Suggestions for Authors

Comments manuscript Tarpgaard et al. Cancers

This concerns a very comprehensive and detailed review of developments in the treatment of metastatic colorectal cancer beyond 2nd line. My comments are as follows.

1.      The use of the term “Refractory” in the title, abstract and manuscript is misleading, since this depends on the history of development of late-line treatments. Terms like Late-line or Beyond second line better defines the topic of this manuscript.

2.      Table 2: what is “Danish Lonsurf”? This acronym is not used in the publication of this trial.

3.      Discussion: The sentence “If the tumor is RAS and BRAF wild-type you should check that the patient did receive prior anti-EGFR therapy and that patients with dMMR and BRAF mutated tumors have been exposed to immunotherapy and BRAF inhibitors, respectively. If not, the patient should of course be offered the relevant treatment” requires editing, since 1) tumor sidedness should be included in patients with RAS/BRAF wt tumors, 2) it suggests that these patients may also receive anti-EGFR in 3rd line, which is generally considered as bad practice, and 3) it should be dMMR or (and not and) BRAF mutated tumors, as well as BRAF V600E instead of only BRAF.

4.      My main comment is that, with only few exceptions, the manuscript lacks recommendations for the use of the drugs that have been discussed. Patients usually do not live long enough to be exposed to all currently available options, so this review should also aim to guide medical oncologists in the choice of treatment. For instance, when no specific target is known or available, is TTP or regorafenib the preferred drug? Should TTP + bevacizumab be the new standard? Should patients who are candidate for 3rd line treatment be tested for specific mutations? If such a target is present, should targeted treatment be preferred over non-specific treatment? Which patients with RAS wildtype (and BRAFV600E wildtype should be added here!) tumors are candidates for rechallenge with anti-EGFR? To recommend that these patients should be treated with anti-EGFR plus irinotecan based on the BOND trial may be questioned since no data on RAS/BRAFV600E status are available from this trial.

Author Response

Dear Reviewer 1,

We thank you for your very useful and important comments and suggestions for our manuscripts. We have addressed them all in the attached file.

We have made a REVISED version with track-changes, and a CLEAR version of the manuscript.

We hope these changes have made the manuscript ready for publication and look forward to hear from you again.

Yours sincerely

Line Schmidt Tarpgaard

Reviewer 2 Report

Comments and Suggestions for Authors

The authors of this review aim to present the treatment options for patients with chemorefractory metastatic CRC based on randomized trials and to analyze their efficacy and tolerability.
Based on the analysis of studies including unselected patients, the characteristics of CRCs with a molecular biological background (e.g., BRAF mutations, KRAS G12C mutations, HER2 amplification, deficient 25
MMR, or NTRK gene fusions) are summarized in a concise and understandable way.
Second-line treatment options are discussed in light of the recommendations.
Special attention is given to studies where the progression-free survival of patients was not found to be beneficial with the therapies used.
The review is well structured, its objective is clear, and the results presented are clear. The summary tables are also clear and illustrative. The literature used is up-to-date and adequate.
The use of English is appropriate.
The length of the manuscript is somewhat short; I recommend it for acceptance as a mini-review.

Author Response

Dear Reviewer 1,

Thank you for your comments.

We were asked for a 4000-word text.

This can be categorized as a mini-review.

Best regards, 

Line Tarpgaard